# Sustainable Campus: The Experience of the University of Lisbon at IST

**João Gomes Ferreira** [1,*] , **Mário de Matos** [2] , **Hugo Silva** [2] , **Afonso Franca** [2] and **Pedro Duarte** [2]

1   CERIS, Instituto Superior Técnico, University of Lisbon, 1049-001 Lisboa, Portugal
2   Instituto Superior Técnico, University of Lisbon, 1049-001 Lisboa, Portugal;
    mario.matos@tecnico.ulisboa.pt (M.d.M.); hsilva@tecnico.ulisboa.pt (H.S.);
    afonso.franca@tecnico.ulisboa.pt (A.F.); pedrolimaduarte@tecnico.ulisboa.pt (P.D.)
*   Correspondence: joao.gomes.ferreira@tecnico.ulisboa.pt

**Abstract:** This paper addresses the research problem of determining if relevant energy and water savings may be obtained in university campuses without significant investments, based mainly on "surgical" technical and organizational measures. With the creation of the "Sustainable Campus" project, in 2011, IST has been implementing a sustainability policy. A resource management policy has been accompanied by a permanent project team, which proposes the implementation of technical measures. This activity is performed in articulation with the operational management through integrated actions in a collective effort to reduce consumption. Without significant investments, the energy-saving measures implemented have consistently achieved a value of more than 20% in energy consumption when compared to the average annual consumption values of the past decade. Additionally, in 2018, water consumption was 58% lower than the reference baseline value of 2011. In 2018, specific projects were implemented in the area of sustainable mobility, with a focus on smooth mobility and sharing. This paper presents the "Sustainable Campus" project, its operational lines, and the results achieved in energy and water consumption and sustainable mobility.

**Keywords:** sustainable campus; energy efficiency; hydric efficiency; resource management; sustainable mobility





## 1. Introduction

The limits of the Sustainable Campus concept are difficult to establish in a concise and definitive way [1–5]. The wideness of the sustainability concept, particularly when extended to its social and economic components, makes it difficult to characterize it based only on rigorously quantifiable and measurable indicators. However, environmental sustainability specifically associated with the use of resources is based on high rigor and a rational and careful use of those resources, leading to high levels of energy efficiency and hydric efficiency in the activities carried out at the university campuses.

Besides energy and water consumption on the campuses, it should be noted that the daily life of the members of the academic community also implies a set of journeys only possible, in the great majority, by means of motorized mobility. Thus, issues related to mobility must also be included in the programs to improve the sustainability of campuses [6] whenever the academic community includes members from surrounding areas that may (or must, in practice) use motorized mobility, as is the case of the IST.

IST, the largest School of Engineering, Architecture, and Technology in Portugal, has three campuses. The Alameda campus is located in the center of Lisbon and has a ground area of about 110,000 $m^2$. The Campus Tecnológico Nuclear (CTN) is located in the municipality of Loures and presents a ground area of about 20,000 $m^2$. Finally, the Taguspark campus, in the municipality of Oeiras, which includes the most recent IST building, was completed in 2009, with 30,000 $m^2$ of ground area. In the metropolitan area of Lisbon, IST also has three university residences.

In general, IST's facilities at the various campuses consist of independent buildings in which training, research and development, administrative activities, and other complementary support activities are carried out. The buildings are distinguished by the type of activities and by the nature of the engineering fields developed there but also by the different construction periods. Buildings from the first half of the twentieth century coexist with buildings whose construction ended in the early years of this century. This diversity is also associated with an effective difference between the needs and constraints in the management of resources within these buildings.

About 90% of the people who access IST campuses on a daily basis are students. This population has been increasing since 2004/05 and then approximately stabilized in the last 5 years, as shown in Figure 1. In 2004/05, 7719 students were enrolled on Alameda campus, while in 2017/18, that number raised to 9869 [7]. In the same period, the number of students enrolled in Taguspark campus also increased significantly.

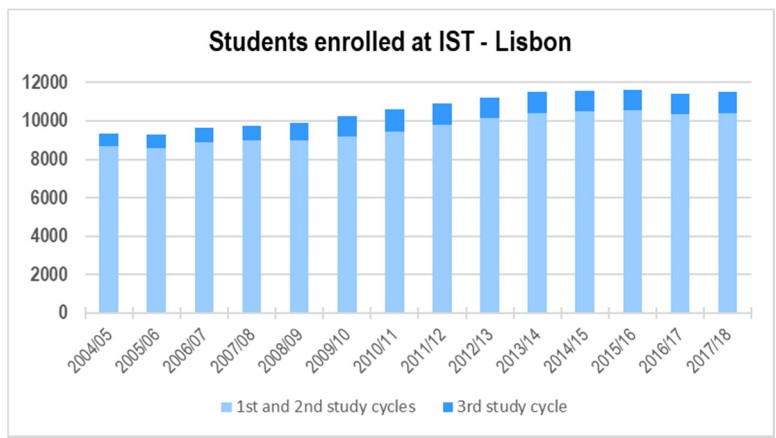

**Figure 1.** Evolution of the number of enrolled students at IST.

In the academic year 2017/18, 11,533 students were enrolled, and around 869 faculty members and 552 technical and administrative staff members were employed, totalizing 12,954 academic community members. This means that the campuses of IST move a number of users that is higher than the resident population of more than half of the Portuguese municipalities.

From the end of 2011, IST has been gradually implementing a policy of sustainability at a central level, with the creation and promotion of the "Sustainable Campus" project. This project, initiated by the Executive Board in 2012, was conceived with a view to the concrete implementation of tangible actions in the IST technical systems and practices and the encouragement of enhanced behaviors of community members in daily life at the campuses. A detailed description of the fundamentals for choosing the model initially adopted, as well as of the first years of the project, was presented by Ferrão and Matos [8]. The objective of the project was to position IST as a reference institution in university Sustainable Campus policies [9], reinforcing its role as a social development agent among students and society in general.

## 2. The Operation of Sustainable Campus in the IST Organization

The "Sustainable Campus" project was conceived with the objective of improving the efficiency in the use of resources in all IST installations, increasing the energy and hydric performance of the campuses, and simultaneously consolidating a reinforcement of skills in the use of resources and sustainability in the academy.

On one hand, the objective of the project implementation is to place IST as a reference in Sustainable Campuses, with significant results in terms of reducing its own energy and water bills and the consumption of other resources. On the other hand, it is intended to involve the entire academic community of students and faculty members through practical

application, leveraging existing knowledge on subjects related to resource efficiency, sustainable mobility, waste management, and the overall implementation and management of a sustainable campus.

In the beginning, and over the period between 2012 and 2015, the project had a dedicated coordinator and was supervised by the leader of the IST transversal platform called "IST Energy Initiative". Since the beginning of 2016, the coordinator of the "Sustainable Campus" project has been working directly with the IST Executive Board through the Vice President responsible for the Infrastructure Management Service.

At the operational level, the project, since its beginning, has had a dedicated team that proposes and implements concrete actions, regarding the technical systems and the daily life of the campuses, constantly performs the tasks of energy management and supervision of water use, and serves as the link between the various actors in these matters. These actors include students, faculty, and technical and administrative personnel, as well as all other users of campuses. With autonomy and independence, the Sustainable Campus team contributes technically to the decision-making process in maintenance, works, and mobility.

Supported by a central policy of sustainability, directly supervised by the Executive Board, the "Sustainable Campus" team works closely with the Infrastructure Management Service of the IST, through integrated actions with the respective Nuclei of "Maintenance", "Works", and "Safety, Hygiene, and Health", as well as with the managers of the campuses buildings. In this context, a collective effort to reduce the consumption of resources has been maintained through a more rational and efficient operation and maintenance of equipment and a progressive introduction of sustainability criteria in the design of new infrastructures, in the rehabilitation of facilities, and in procurement.

Up to now, practical studies and works have been developed in the context of the Sustainable Campus by more than a hundred students, supervised by about fifteen professors from different departments. Among these works are 25 Master's theses and numerous practical classes in the field of energy efficiency, rationalization of water use, and sustainability on university campuses. The "Sustainable Campus" project also cooperated directly with five other R&D projects in which IST is currently, or was previously, involved.

Because of the high number of participants but also their transversal reach of the entire academic community, the projects in the area of Sustainable Mobility already developed resulted in a greater visibility of the Sustainable Campus and, consequently, an increase in collective awareness of issues related to sustainability.

One of the main objectives of the Sustainable Campus and of the Infrastructure Management Service is to extend these good practices and awareness [10,11] to the field of waste management, reuse, and recycling for all elements of the academic community.

## 3. Applied Methodologies

### 3.1. Energy Consumption

The initial work on the characterization of the energy consumptions involved the completion of 43 exhaustive Energy Audits in the three campuses, which included energy surveys in the outdoor systems and diagnoses to the main energy equipment [12–14]. These audits led to the study and evaluation of dozens of energy efficiency measures, adapted to each building, which were categorized by levels of investment [15,16].

Historical average values show that energy consumption on Alameda's campus represents three-quarters of IST's energy bill, of which only a share of less than 2% corresponds to the actual costs associated with natural gas, consumption being the remaining 98% of electricity costs. Therefore, the consumption of electric energy in the campus of Alameda has deserved special attention since the beginning of the "Sustainable Campus" project. Before the start of the project, between 2006 and 2010, annual consumption was characterized by values of around 14 GWh/year, as shown in Figure 2, with a daily minimum supply power of 1 MW, to which all the daily operational activity of the campus was added. During that period, the natural gas consumption was around 120 (n)km$^3$/year.

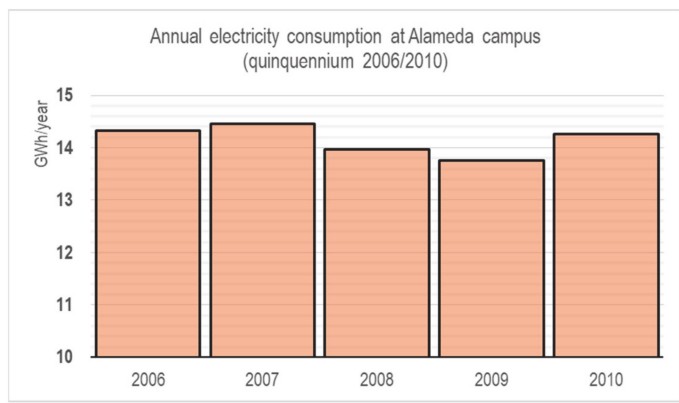

**Figure 2.** Energy consumption at Alameda campus, prior to the "Sustainable Campus" project.

Considering the constructed area of the Alameda campus of 110,000 m$^2$, 14 GWh/year corresponds to a consumption of about 127 kWh/m$^2$/year, and the specific natural gas use was then 1.1 (n)m$^3$/m$^2$. For comparison, the average values in colleges and universities in the USA were around 204 kWh/m$^2$/year for electricity plus 5.2 (n)m$^3$/m$^2$ of natural gas consumption [17]. In Germany, in terms of primary energy use, Leuphana University Lüneburg reports a consumption of 146 kWh/m$^2$/year, prior to the installation of a photovoltaic system, whereas the university complex of the TU Braunschweig, comprising buildings built from 1918 to 2007, reports 293 kWh/m$^2$/year prior to the implementation of saving measures [18]. Regarding the 2018 UI GreenMetric World University Ranking [19], the first university ranked in the Europe region, the Wageningen University and Research in the Netherlands, reported for the year 2017 values of 112 kWh/m$^2$/year for electricity plus 12 (n)m3/m$^2$ for natural gas in their annual environmental report [20], and the second ranked, the University of Nottingham in the UK, reported for the year 2017 in its annual energy report [21] a value of 300 kWh/m$^2$/year for the total energy consumption (electricity and fossil fuels), which corresponds to a value of 5963 kWh/student/year.

It must be taken into account that the electric energy consumption may vary significantly between campuses, depending on their specific characteristics, such as location/climate, building age, passive behavior (insulation and solar gains), technical management, renewable energy sources available, type of equipment installed, etc. In the case of IST, the energy consumption values regarding the period between 2006 and 2010 may not seem quite significant. However, the analysis made in 2012 and in the early years, in the context of the beginning of the "Sustainable Campus" project, clearly showed that the electric energy was not being efficiently used, in general, as would be proven by the results achieved later. Some of the reasons for this apparently low value, below 130 kWh/m$^2$/year for the consumption in those years, comparatively with the indicators referred above from other universities, are mostly related to the favorable climate in Lisbon and the scarcity of HVAC systems, which leads to a lack of comfort. Nevertheless, it should be noted that this figure included some buildings in the campus that have a consumption of nearly 230 kWh/m$^2$/year, as is the case of the South Tower, and others with a per square meter consumption as low as 100 kWh/m$^2$/year, such as the Civil Pavilion.

Several energy-saving measures, with negligible levels of direct investment, feasible by the operational management of the campus, were immediately implemented. These measures included dissemination and awareness-raising for best practice, involving building managers, maintenance technicians, and other employees, who received specific guidelines on energy-saving procedures. Other energy conservation measures included, for example, the establishment of stricter control routines in the equipment's operating hours, in coordination with the IST Maintenance Nucleus.

Periods of reduced activity were defined, with specific rules to avoid unnecessary consumption of energy and water, as well as a progressive centralization of servers and data centers in appropriate facilities, and control and inspection of individual HVAC systems on

campuses. Increased attention was given to unnecessarily functioning equipment, calling for continuous real-time monitoring of energy use.

In this context, meters were placed in installations where no consumption was measured before, and a plan of regular readings of energy and water consumption per building was consolidated. This task allowed the initiation of a rigorous evaluation process of the specific energy, carbon indicators, and other energy-related key performance indicators of each building and to initiate an energy and water consumption management policy based on permanent monitoring.

The automatic tool, originally designed as an energy monitoring system, that was developed for this purpose by a research center associated with IST, consists today of a monitoring platform, which is capable of reading and presenting the information received from sensors spread throughout the Alameda campus and other campuses and IST residences of IST. This platform, called "EnergIST", was customized to allow more granular monitoring in each building as new sensors are being added to the system, as well as the extension to other campuses and the measurement of other physical quantities associated with the consumption and use of resources. The EnergIST interface with the user is shown in Figure 3, where a North Tower HVAC dashboard is presented with the July 2019 readings of the energy consumption associated with the central HVAC system in the building (clear blue line) and the corresponding hourly costs in Euros (black diagram). The platform was described by Kuipers et al. [22] and currently has 74 independent points of reading and an acquisition rate of one reading per minute. It comprises measuring 26 buildings at Alameda campus, the bigger HVAC central systems of the main buildings, and all the concessions associated with contracts made with external companies exploring activities in the Alameda campus, such as bars, restaurants, or even banks. All are being monitored for energy consumption. In addition to that, the platform is connected to the supply points of natural gas to the Alameda campus, to the weather station of the campus, and to the solar thermal system of the IST university residence (Residência Duarte Pacheco), which is located in another place in Lisbon.

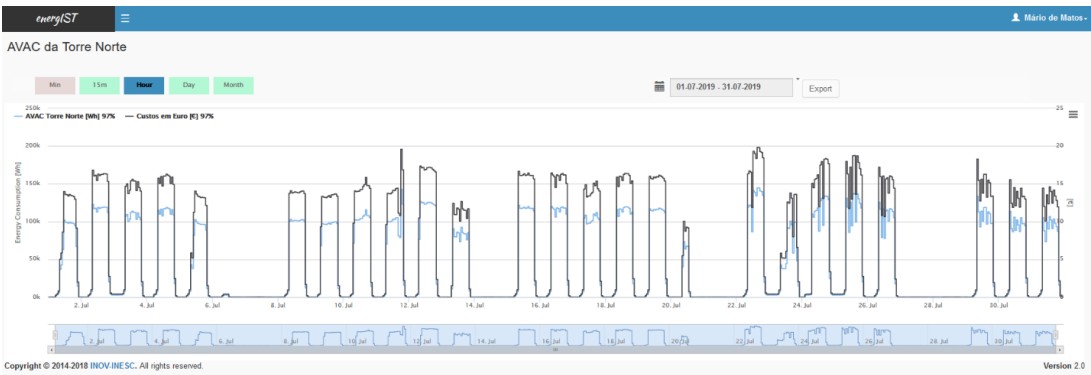

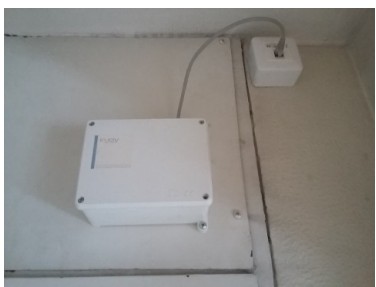

**Figure 3.** "EnergIST" platform: Interface with the user (**box above**); the modbus gateway receives information from the meters and sends it to the IST computer network (**box below**).



Campus buildings' water monitoring is now in a test phase, with the installation of new flowmeters with pulse sensors, and will be connected to the EnergIST platform during the next year.

### 3.2. Water Consumption

Prior to the implementation of the "Sustainable Campus" project, some routines were carried out by the security company personnel to read the water consumption in the buildings. However, an integrated monitoring and control strategy was not implemented. In particular, the summation of local measurements was not performed, and no comparison between those measurements and the water bill was carried out. From the beginning of the year 2012, these routines were implemented, and data regarding the previous two years were retrieved.

In 2010, water consumption on Alameda's campus was 140,000 m$^3$/year, considering the three supply points from the public mains water corresponding to 85% of water consumption in the total of the IST's facilities. This annual consumption would fill about 56 Olympic-sized swimming pools, with $50 \times 25 \times 2$ m$^3$. This water consumption corresponds to an annual average of over 30 L/day/campus user or 33 L/day/student.

A deeper look at these numbers and the large amount of water they represent clearly pointed out the need for greater efforts to understand the state of the hydric efficiency on this campus.

However, the water consumption on university campuses may vary significantly. Besides the local rainfall, it depends on the kind of gardens and green areas, their size, or whether or not there are university residences inside the campus, and naturally, the type of activities carried out inside the buildings related to the courses given and to the research practices. For example, the water consumption in the campuses of the University of California varies from values of a 2015 annual average of 75 l/day/weighted campus user (WCU) in Santa Barbara [23] to values of 170 l/day/WCU with a target of 155l/day/WCU in the campus of Los Angeles [24]. In Malaysia, the University Teknologi Malaysia consumed 260l/day/campus user for the year 2002 [25] and, for Poland, Wichowski et al. [26] analyzed in detail the registered water consumption in years 2012 to 2016 at the Warsaw University of Life Sciences, WULS, determining 26.6 l/day/student for full-time students for the year 2016 as the annual average consumption. Regarding the 2018 UI GreenMetric World University Ranking [19], the first two universities ranked in the Europe region, the Wageningen University and Research in the Netherlands and the University of Nottingham in the UK, have reported for the year 2017 values of 35 l/day/campus user and 65 l/day/student in their annual environmental report [20] and annual energy report [21], respectively.

Based on management practices supported by the permanent monitoring of water consumption, associated with a careful maintenance of the distribution networks inside the campuses and an immediate intervention in case of any rupture or a significant deviation from the expected pattern of consumption, a very sharp reduction in water consumption was then observed, avoiding any abnormal loss or excessive consumption.

Low-value investments supported by the current management budget of the IST allowed several "surgical" but high-impact repairs to be made, which allowed the elimination of important old leaks in the water distribution network inside the campus.

The most significant examples of water waste precisely regard these old pipes of the water supply system in Alameda. At some points, there were severe ruptures leading to significant leakages that were not detected because no water emerged at the ground surface and no monitoring was carried out. The water waste on water supply leakages was estimated at more than 20,000 m$^3$/year. Other diverse situations were identified, with different waste values, such as a case where new refrigerating water was permanently being consumed by laboratory equipment where the system was conceived to recirculate the cooling water or a defective float water valve of the hydraulic laboratory flow channel.

Routine inspection by the IST Safety and Health personnel at the end of the daily activities on the campuses, and the subsequent immediate report to central coordination, has enabled the timely correction of situations of waste due to misuse or neglect.

### 3.3. Sustainable Mobility

Throughout 2017, two specific projects were developed in the area of sustainable mobility for the users of the IST campuses. The projects naturally have a very strong focus on promoting modes of transport associated with smooth mobility and sharing: a carpooling platform restricted to the IST academic community and the project "U-Bike Portugal—Operation Tecnico" (Figure 4).

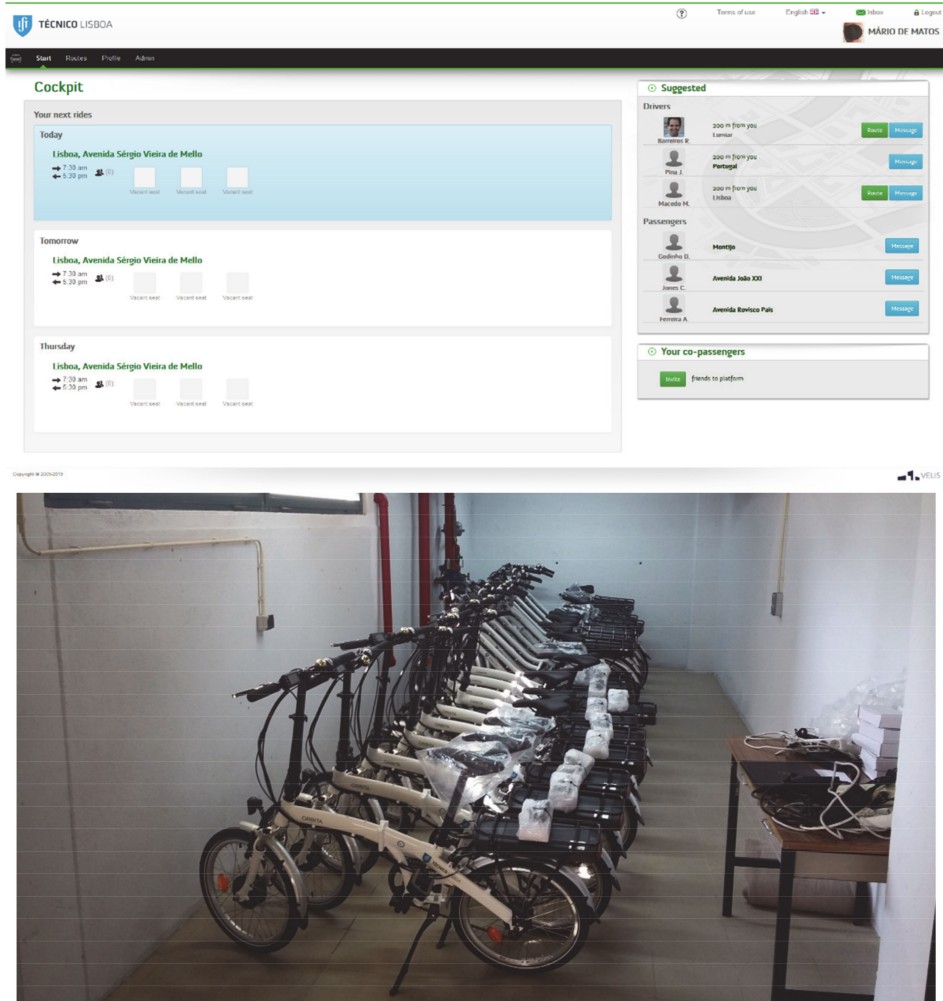

**Figure 4.** Sustainable mobility projects: user interface of the IST's exclusive carpooling platform (box above), IST bicycles for use by the academic community (box below).

At the beginning of the 2017/18 school year, IST launched an exclusive carpooling platform to serve the academic community (https://carpooling.tecnico.ulisboa.pt accessed on 8 January 2020), allowing the sharing of traveling by car for those who prefer to use this means of transport to travel to, from, or between IST campuses. The potential reduction of the total costs of transport by car and corresponding reduction of emissions can reach 75% in the case of four people sharing one single common vehicle. The cost associated with car parking in the area surrounding the Alameda campus can also be reduced, and parking problems inside the campuses are minimized.

The users of the platform may assume two roles: the driver and the passenger. The driver–user specifies his route, days of travel, possible itinerary points, and time of the

meeting for the hitch. Passenger users check the availability of travel according to their travel needs and choose their travel through the platform.

Based on the users' indications, the platform suggests nearby trips that may correspond to those who are hitchhiking or want to share a trip in the future. Whenever users find a correspondence between interests and availabilities, a direct contact is made between the driver and the passenger in order to arrange the details of the hitch: the exact time, place of meeting, any stops, or cost sharing.

During the COVID-19 pandemic, the system has been quite less attractive and, in practice, was suspended, because carpooling involves an actual risk of contamination. Moreover, during the periods when the emergency state or calamity state was declared, it was even illegal to use it. There is an expectation that the adhesion to the system may increase after the pandemic, especially because a global saving attitude—saving money, the environment, and resources in general—has grown all over with the crisis. Nevertheless, after the pandemic is over, a promoting campaign will be carried out in IST to disseminate this tool and let the community become more familiarized and comfortable in using it.

In promoting a more sustainable mobility of the academic community, a particular focus was also given to the electric bicycle as a preferential means of transport in daily access to the IST campuses. For this purpose, the project "U-Bike Portugal—Operation Tecnico", financed by the EU through the Portugal2020 program, aims at disseminating more sustainable urban displacement practices.

Within the scope of the project, a set of 20 electric bicycles for own use during periods of one semester was made available to the academic community, mainly to students, at affordable costs. Electric bicycles were adopted because they allow for longer and more sloped rides and thus may be adopted by a larger number of community members. In particular, they may be adopted by community members who live further away and normally use more polluting and less healthy transportation, allowing the accomplishment of the objective of substituting motorized rides. Bicycles are allocated by lottery, which takes place at the beginning of each school semester.

The project included the construction of new parking areas for bicycles on the three campuses of IST, the installation of a self-service station at Alameda for backing and small repairs, and the implementation of a communication plan for the dissemination of bicycles as a sustainable and healthy means of transportation.

This project intends to be an incentive to use bicycles as a preferential means of transportation in daily access to the IST campuses, functioning as a catalyst for a tendency to substitute the motorized vehicle with the bicycle. It was expected to achieve a 50% increase in the use of the bicycle as a means of preferential travel for the campuses until 2020. This corresponds to a number of 350 to 500 people, mostly students, in the class period, going to IST daily by bicycle.

Of course, the pandemic has had a strong impact on academic life, with most of the classes happening online and significantly less home–school traveling occurring.

At the same time, while IST was implementing the U-Bike program, the Lisbon municipality was developing a low-cost bike-sharing program, called GIRA, that is presently implemented. This program has involved the construction of more than one hundred kilometers of bike lanes and has made available about 1500 bicycles and 200 docks. The program, with a massive adhesion by Lisbon citizens, especially among young people, has strongly complemented the IST U-Bike project for home–school traveling, with several bike dockings installed near the diverse campus entrances.

Between U-Bike and GIRA, each program has its advantages. The GIRA project has the advantages, for example, of being widespread all over the city and not compromising the user with the bike maintenance. On the contrary, in the U-Bike program, the user is responsible for the bike's maintenance but enjoys its exclusive use with permanent availability and is not constrained by docking locations, benefitting from door-to-door traveling.

## 4. Results Achieved

### 4.1. Energy Consumption

The energy conservation measures implemented in the course of the "Sustainable Campus" project on the Alameda campus have already stabilized a persistent saving of about 16% when compared to the year 2011. The year 2011 was assumed as the baseline year for accounting of the results of the project since it was the year preceding its start and nevertheless, it presented a consumption lower than the previous quinquennium, as confirmed by comparing Figure 2; Figure 5.

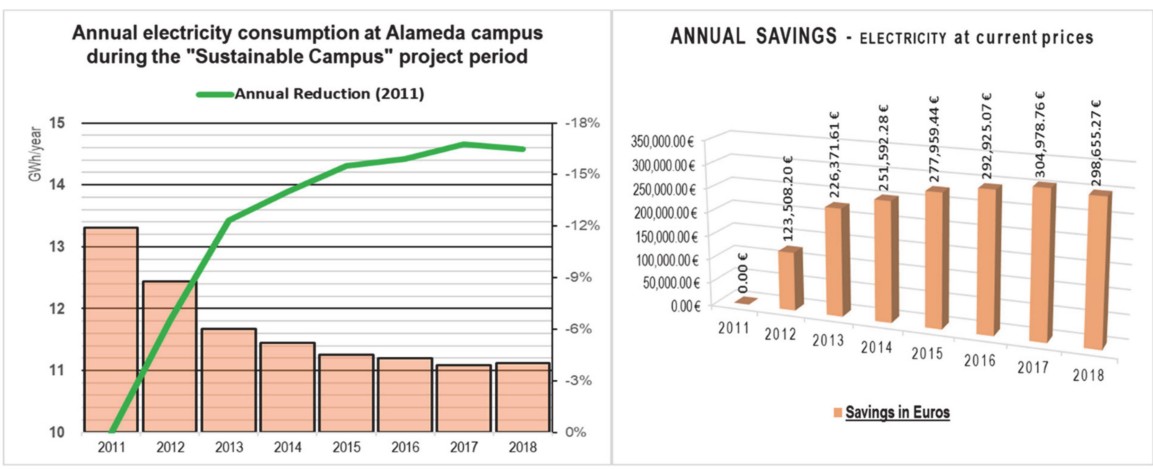

**Figure 5.** Evolution of energy consumption at the Alameda campus, after the implementation of the "Sustainable Campus" project (baseline of 2011) (**left-hand side**); annual costs avoided with the electricity bill (**right-hand side**).

Figure 5 shows the annual costs avoided in the electricity bill on the campus of Alameda after the implementation of the "Sustainable Campus" project at IST, with reference to the consumption baseline of 2011. In each year, the total unit price (blended cost per kWh, i.e., total value of the bill per kWh, including taxes and other aggregated costs) is applied to the consumption differential versus 2011, determining the saving in the annual bill.

In fact, despite an average increase of 20% in tariff costs between 2011 and 2018, the electricity bill for the Alameda campus, in 2018, without any loss of service, lowered by more than 250,000 euros when compared to the invoice for the year 2012 (the first full year with VAT on electricity at 23%).

Annual variations of less than 1%, consecutively in the last 3 years, as shown in Figure 5 (left), show that the energy consumption in this campus is controlled and the results achieved in the improvement of campus energy performance are consolidated. The value already achieved corresponds to less 1135 kWh/person/year (~11.2 GWh/year/9869 Alameda students), i.e., a permanent (24 h/24 h) power of 130 W/student, which is satisfactorily aligned with other institutions with sustainability concerns and good practices already implemented for a longer period.

Since the beginning of the project, taking as reference the baseline of energy consumption of the year 2011, a total amount of 13 GWh in electricity was saved at the Alameda campus. This saving is equivalent to 2791 tonnes of primary energy and represents 6102 $tCO_2eq$ of avoided GHG emissions.

The significant reductions in energy consumption were obtained based on a combination of actions. Firstly, annual periods of reduced activity (15 consecutive days in August) were defined, when all activities were interrupted or reduced to a minimum and most employees took holidays. This already allowed for a certain energy saving, but most of all, it allowed defining the consumption baseload values and identifying the equipment responsible for it. Some of this equipment had to be functioning in the current regime, but some was functioning during unnecessary periods, causing energy waste. This included

a variety of equipment, mostly HVAC and illumination that, since then, was turned off during the unnecessary periods. This approach allowed reducing the baseload from about 1000 kWh/h to about 700 kWh/h, corresponding to about half of the overall energy saving. The other half of saving was due to diverse measures, namely: implementation of central real-time monitoring and control systems, allowing identifying and turning off equipment during unnecessary periods; definition of more rational routines in the equipment's operating hours; centralization of servers and data centers in appropriate facilities with dedicated refrigerating systems; implementation of control and inspection routines of individual HVAC units; more rational use of laboratory equipment; and spontaneous saving gestures, such as turning off the light of the heater when leaving the office, as a result of an increased awareness among the community regarding the environment and energy waste.

### 4.2. Water Consumption

The evolution of the annual water bill evidences a very marked progressive reduction in consumption on the Alameda campus since 2011. Total water consumption on the campus of Alameda in 2018 was 58,359 m$^3$/year, contrasting with values of about 140,000 m$^3$/year at the beginning of the decade, as observed in Figure 5. This means that by 2018, compared to the reference baseline of 2011, which, as in the case of energy, was adopted to account for the results of the "Sustainable Campus" project, the total reduction achieved was 58%.

Figure 6 shows the annual costs avoided in the water billing of the Alameda campus after the implementation of the project and the methodologies that were described in Section 3.2. The cost per cubic meter of each year is applied to the difference between the respective annual consumption and that of 2011, obtaining the value of the savings. For the year 2018, the annual saving in the bill associated with the reduction of consumption was over 295,000 Euros. Taking the baseline of 2011, the water savings accumulated since the beginning of the project correspond to 387,426 m$^3$, meaning 155 Olympic size pools (50 × 25 × 2 m$^3$) and a financial value of more than one million Euros.

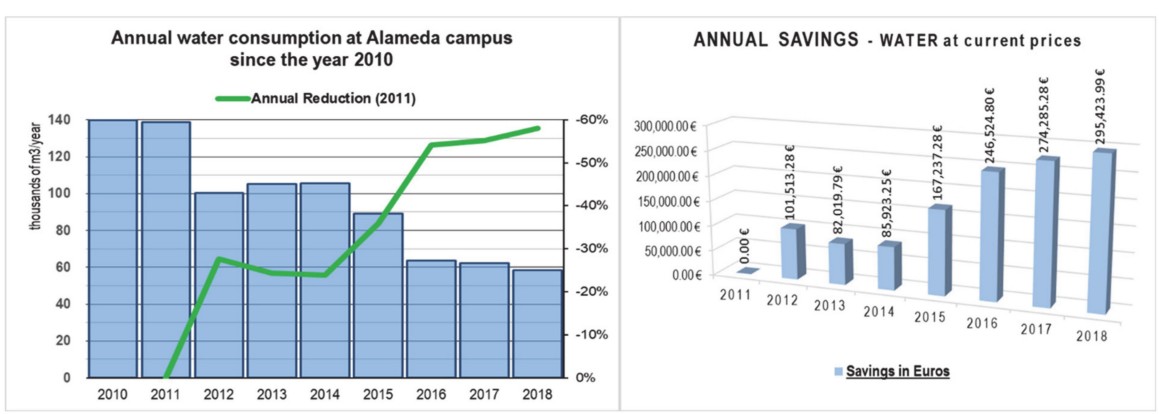

**Figure 6.** Evolution of water consumption at Alameda campus, after implementation of the "Sustainable Campus" project (baseline 2011) (**left-hand side**); Annual costs avoided with water bills (**right-hand side**).

In the last 3 years, the water consumption continues to decrease, but at a less pronounced rate. On the one hand, this fact shows that it can still be improved. On the other hand, the stabilization trend seems to signal the approximation to a referential level of water efficiency of the campus, which is estimated to be close to 50,000 m$^3$/year under current conditions, as a more detailed analysis of the water consumption daily profile appears to point out. This corresponds to about 14 L/person/day (considering the 9869 Alameda students enrolled in 2018).

### 4.3. Sustainable Mobility

The operation of the project "U-Bike Portugal—Operation Tecnico" began in the first semester of the academic year of 2018/19. The interest of the academic community in the project was demonstrated by the considerable number of registrations made, which exceeded by more than 10 times the available bicycles on offer.

The first lottery for assigning the 20 bicycles of the project took place in December 2018. Until August 2019, about 19,698 counted kilometers were already traversed by members of the academic community, including students, faculty, and technical and administrative personnel. The users of bicycles participate actively in the gaming challenges that IST promotes in the scope of the project, having reached marks such as 900 km traveled by a single bicycle, in the first quarter of 2019 during the winter, of which almost 700 km was electrically assisted.

In the scope of the project "U-bike Portugal—Operation Tecnico", new parking areas were installed on the campuses of Alameda, Taguspark, and CTN exclusively for bicycles. The demand for these zones, both from users participating in the project and users with their own bikes, has been so high since the beginning of the second semester of 2018/19 that there are periods of the day when the parking structures installed on the Alameda campus are full, showing a need for its extension in the short term.

The cycle workshop, created under "U-bike Portugal—Operation Tecnico" to assist the participants in the project, also serves the entire cycling community of IST with mechanical services of minor repairs. In its exterior, but in an environment protected from the weather, a full self-service station is installed, permanently open, which provides a continuous service, releasing bicycle users from the rigidity of office hours when a mechanical emergency arises.

Table 1 presents the indicators regarding the participation and use of the IST exclusive carpooling platform in the first twenty months of operation.

**Table 1.** Indicators of adherence to IST's carpooling platform (September 2017 to May 2019).

| **Number of Registered Members** | **1017** |
|---|---|
| Number of registered motor vehicles | 310 |
| Number of drivers with travel offers | 69 |
| Number of itineraries with different routes and times | 82 |
| Total number of trips offered | 20,279 |
| Kilometers traveled with effective transportation offer | 832,104 |
| Total number of hitchhiking places available | 62,856 |
| Number of available travel kilometers | 2,642,414 |

Table 1 shows that about 8% of IST academic community members have already registered for the platform. These indicators allow us to consider the potential for fuel savings associated with the use of the platform, which could reach 210 thousand liters, representing reductions of GHG emissions of up to 340 tCO$_2$eq and more than 300 thousand Euros in fuel.

The coincidence of interests at equal times and journeys over small distances, such as daily trips to the IST campuses, is one of the greatest difficulties in the widespread adhesion to this mode of transportation. Thus, despite the high interest shown in the registration phase, the implementation of hitchhiking, assessed based on platform registrations, is still small, leaving a high potential for growth. It should be noted that the platform allows a first contact between people but does not require that the communications between them continue to take place there. Thus, once they have met, drivers and passengers can arrange shared travel by direct contact. In these cases, shared travel is not counted, but the initial goal of sharing is achieved.

## 5. Future Developments

Regarding the efficiency in the use of resources, it is intended to maintain the current policy of energy management and monitoring and control of water consumption, reinforcing and optimizing the measures that were being implemented throughout the "Sustainable Campus" project, with the objective of persistently stabilizing the savings values achieved. The characterization and survey of the technical systems of the CTN campus are being finalized, aiming at initiating the Energy Certification process for this campus, which is an essential condition for applying to national and EU funding.

Currently, IST is already implementing a very relevant plan for energy efficiency on the Alameda campus, named the "Energy Efficiency Plan—Tecnico 2020", funded by the EU through the national POSEUR—Operational Program for Sustainability and Efficiency in the Use of Resources in the scope of Portugal2020. This project consists of several measures for energy efficiency improvement that will be implemented between 2019 and 2021, with an overall investment value of around 5 million Euros. These measures comprise interventions in HVAC systems installed to increase energy efficiency, replacing indoor lighting with LED systems, and incorporating solar thermal energy and photovoltaic on campus for self-consumption. All eight measures to improve energy efficiency are included in the Energy Assessment Report, prepared following the Energy Audits to the Alameda campus carried out by the project team of the "Sustainable Campus" project. With the implementation of the "Energy Efficiency Plan—Tecnico 2020", an additional energy saving of 21% is estimated for the Alameda campus. The plan implementation will involve significant modernization and rehabilitation investments, which will increase considerably the comfort of campus users. The implementation of all the interventions foreseen in this plan to improve energy efficiency will result in the achievement of the national targets established for public administration to achieve a reduction of 30% in consumption by 2020 compared to 2007 baseline values.

Concerning water efficiency, in order to achieve further improvements, it will be necessary to ensure greater investment capacity to modernize distribution networks and to install flow control systems and monitoring equipment. Several low-cost measures to be progressively implemented are foreseen for the next 3 years, involving technical and management measures as well as an awareness increase through communication that will allow reducing the total water consumption at IST by more than 10%. One of the measures to be implemented consists of the reuse of water for the irrigation of gardens.

With regard to sustainable mobility, a new project for the installation of charging stations for electric vehicles in eight parking places has been completed and is now only waiting for budgetary availability during the current year. With this new infrastructure, planned for the Alameda campus in this initial phase, IST will enhance conditions to increase the electric mobility of members of his academic community. In the area of sustainable mobility, it is also intended to encourage more consistently the sharing of hitchhiking through the carpooling platform by increasing and improving the efficiency of communication actions. Additionally, it is intended to increase the number of electric bicycles available for school semester distribution, with the support of specific public funding.

Waste management is currently the responsibility of the IST Safety, Hygiene, and Health Nucleus. One of the developments to be implemented is the sharing of this responsibility between this service and the "Sustainable Campus". As with the energy, water, and mobility management aspects, "Sustainable Campus" will monitor the waste management data and propose additional measures to be implemented. In this field, waste management monitoring presents specific difficulties, namely because there are no commercially available solutions for the measurement of volumes of waste that are practical, economical, and reliable and simultaneously integrable in existing systems. It is therefore intended to develop tools for the monitoring of quantifiable parameters in the field of waste management and recycling in collaboration with students, teachers, and researchers. These tools should allow the characterization and monitoring in detail of most parameters regarding waste management and the measurement of the results of the measures implemented.

## 6. Conclusions

The presented case study has contributed to positively responding to the initial research question of determining if relevant energy and water savings may be obtained on university campuses without significant investments, based mainly on 'surgical' technical and organizational measures. The data obtained until 2018 were used when the savings had practically entered a plateau. It was considered that, from that point on, further significant savings required significant investments. The authors prepared and submitted an application to POSEUR, the Portuguese Operational Program for Sustainability and Efficiency in the Use of Resources, financed by the EU. The application was successful and allowed obtaining financing of 5 million euros for a broad program with significant sustainable measures at Alameda campus that have been implemented since 2019.

With a total direct investment of less than half a million euros over the last seven years, the "Sustainable Campus" project, in conjunction with the Infrastructure Management Service of IST, has developed a significant and consequent set of actions in the field of sustainability. These actions allowed, among other realizations, the achievement of significant savings in electricity and water consumption and the implementation of sustainable mobility measures.

In addition to the implementation of technical and management measures, the "Sustainable Campus" has also provided support to carry out work in the field of teaching and research in the different areas in which it carries out its activity.

Not least, the "Sustainable Campus" has fulfilled its mission of increasing the awareness of the academic community towards the global problems of sustainability. This mission is carried out through a science and technology-based approach with practical application and tangible results, which internally serves as an example and sustains good practices and, externally, promotes the cause of sustainability and the role of the IST as a school of excellence in these matters.

It is now intended to extend the activity of the "Sustainable Campus" to the area of waste management and recycling, which also envisages a significant improvement of the results already obtained.

**Author Contributions:** Conceptualization, J.G.F. and M.d.M.; methodology, J.G.F. and M.d.M.; software, M.d.M.; validation, J.G.F., M.d.M., H.S.; formal analysis, M.d.M.; investigation, J.G.F., M.d.M., H.S, A.F. and P.D.; resources, J.G.F.; data curation, M.d.M.; writing—original draft preparation, J.G.F. and M.d.M.; writing—review and editing, J.G.F. and M.d.M.; supervision, J.G.F. and M.d.M.; project administration, J.G.F. All authors have read and agreed to the published version of the manuscript.

**Funding:** No specific funding was obtained to support this investigation.

**Institutional Review Board Statement:** This research received no external funding.

**Informed Consent Statement:** Not applicable.

**Data Availability Statement:** The data presented in this paper is available in data bases at IST servers, with no public access.

**Conflicts of Interest:** The authors declare no conflict of interest.

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
