# Peer review of "Sustainable Campus: The Experience of the University of Lisbon at IST"

_sustainability, doi:10.3390/su13148050_

Round 1

Reviewer 1 Report

Very interesting article about way to sustainable campus. The sustainability policy has been implemented very well in scope of project "Sustainable Campus". The sustainable goals were achieved due reduced energy and water consumptions in the campuses as well as sustainable mobility projects.

Please note that the article must be up-to-date. Here are my comments:

1) Be aware that the units must be written correct way in the complete article. Please use superscript number or sign "^". E.g. "m2" in Line 136 must be written "m2" (better option) or "m^2".

2) Did the Covid-19 pandemic influence the carpooling platform? Give few words on that topic.

3) Chapter 3.3, Lines 301-303 -Is the expected increase in the use of the bicycle as preferential travel option achieved in 2020?

4) Chapter 4.1 - Where are the biggest reductions of energy consumption at the campus and why? A statistical overview of such case would be useful.

5) Chapter 4.2 - Please give example of extreme water consumption at campus and its reason (e.g. special test facilities with higher water use). How big was reduction of water consumption for that extreme case?

6) Chapter 4.3 - What is reason for high percentage of electrically assisted cycling?

Author Response

Journal: Sustainability (ISSN 2071-1050)

Manuscript ID: sustainability-1285716

Type: Article

Title: Sustainable Campus: the experience of the University of Lisbon at IST

Authors: João Gomes Ferreira*, Mário de Matos, Hugo Silva, Afonso Franca, Pedro Duarte

Reviewer #1

Very interesting article about way to sustainable campus. The sustainability policy has been implemented very well in scope of project "Sustainable Campus". The sustainable goals were achieved due reduced energy and water consumptions in the campuses as well as sustainable mobility projects.

  • The authors thank the reviewer for appreciating the work developed at IST.

Please note that the article must be up-to-date. Here are my comments:

1) Be aware that the units must be written correct way in the complete article. Please use superscript number or sign "^". E.g. "m2" in Line 136 must be written "m2" (better option) or "m^2".

  • The authors fully agree. Every unit was duly substituted when superscript applies.

2) Did the Covid-19 pandemic influence the carpooling platform? Give few words on that topic.

  • We thank the reviewer for the pertinent question. The pandemic, in practice, has suspended the use of the carpooling platform.

To clarify this situation, the following paragraph was added:

During the Covid-19 pandemic the system has been quite less attractive and, in practice, was suspended, because carpooling involves an actual risk of contamination. Moreover, during the periods when the emergency state or calamity state was declared, it was even illegal to use it. There is an expectation that the adhesion to the system may increase after the pandemic, especially because a global saving attitude - saving money, saving the environment, and resources in general - has grown all over with the crisis. Nevertheless, after the pandemic is over, a promoting campaign will have to be carried out in IST to disseminate this tool and let the community more familiarized and comfortable in using it.

3) Chapter 3.3, Lines 301-303 -Is the expected increase in the use of the bicycle as preferential travel option achieved in 2020?

  • Thank you for the pertinent question. Actually, because of the pandemic, 2020 and 2021 are being very atypical years in what concerns practically all society activities, including academic life. Nevertheless, or maybe precisely boosted by the pandemic, during this period, the smooth transportation has grown substantially in Lisbon. To show the impact of smooth transportation in Lisbon on the IST program, the following text was added in the end of subsection 3.3:

“Of course, the pandemic has had a strong impact on the academic life, with most of the classes happening online and quite less home-school travelling occurring.

At the same time, while IST was implementing the U-Bike program, the Lisbon municipality was developing a low-cost bike sharing program, called GIRA, that is presently implemented. This program has involved the construction of more than one hundred kilometers of bike lanes and has made available about 1.500 bicycles and 200 docks. The program, with a massive adhesion by Lisbon citizens, especially among young people, has strongly complemented the IST U-Bike project for home-school travelling, with several bike dockings installed near the diverse campus entrances.

Between U-Bike and GIRA, each program has its advantages. The GIRA project has the advantages, for example, of being widespread all over the city and not compromising the user with the bike maintenance. On the contrary, in the U-Bike program the user is responsible for the bike’s maintenance but enjoys its exclusive use with permanent availability and is not constrained with docking locations, benefitting from door-to-door travelling.”

4) Chapter 4.1 - Where are the biggest reductions of energy consumption at the campus and why? A statistical overview of such case would be useful.

  • Thank you for the relevant question. The following test was added in the end of sub-section 4.1, aiming at clarifying that matter:

“The significant reductions in energy consumption were obtained based on a combination of actions. Firstly, annual periods of reduced activity (15 consecutive days in August) were defined, when all activities were interrupted or reduced to a minimum and most employees took holidays. This already allowed for a certain energy saving but, most of all, allowed defining the consumption baseload values and identifying the equipment responsible for it. Some of this equipment had to be functioning at the current regime but some was functioning during unnecessary periods, causing energy waste. This included a variety of equipment, mostly HVAC and illumination that, since then, was turned off during the unnecessary periods. This approach allowed reducing the baseload from about 1000 kWh/h to about 700 kWh/h, corresponding to about half of the overall energy saving. The other half of saving was due to diverse measures, namely: implementation of central real-time monitoring and control systems, allowing identifying and turning off equipment during unnecessary periods; definition of more rational routines in the equipment's operating hours; centralization of servers and data centers in appropriate facilities with dedicated refrigerating systems; implementation of control and inspection routines of individual HVAC units; more rational use of laboratory equipment; spontaneous saving gestures, such as turning off the light of the heater when leaving the office, as a result of an increased awareness among the community regarding the environment and energy waste.“

5) Chapter 4.2 - Please give example of extreme water consumption at campus and its reason (e.g. special test facilities with higher water use). How big was reduction of water consumption for that extreme case?

  • Thank you for the question. Aiming at responding it, the following paragraph was added in sub-section 4.2:

The most significant examples of water waste precisely regard these old pipes of the water supply system in Alameda. In some points, there were severe ruptures leading to significant leakages that were not detected because no water emerged at the ground surface and no monitoring was carried out. The water waste on water supply leakages was estimated in more than 20.000 m3/year. Other diverse situations were identified, with different waste values, such as a case where new refrigerating water was permanently being consumed in a laboratory equipment where the system was conceived to recirculate the cooling water or a defective float water valve of the Hydraulic laboratory flow channel.

6) Chapter 4.3 - What is reason for high percentage of electrically assisted cycling?

  • Thank you for the question. To answer it, the following sentence was added in sub-section 3.3:

“Electric bicycles were adopted because they allow for longer and more sloped rides and thus may be adopted by a larger number of community members. In particular, they may be adopted by community members who live further and normally use more pollutant and less healthy transportation, allowing to accomplish the objective of substituting motorized rides.”

Reviewer 2 Report

"This paper presents the "Sustainable Campus" project, its operational lines and the results achieved in energy and water consumptions and in sustainable mobility." --- This paper presents a case study. However, the purpose of conducting this case study is not at all clear. What is research problem and hypothesis of this study?

I suggest re-writing of abstract and conclusion to clearly highlight research problem, hypothesis and contributions.

Author Response

Journal: Sustainability (ISSN 2071-1050)

Manuscript ID: sustainability-1285716

Type: Article

Title: Sustainable Campus: the experience of the University of Lisbon at IST

Authors: João Gomes Ferreira*, Mário de Matos, Hugo Silva, Afonso Franca, Pedro Duarte

Reviewer #2

"This paper presents the "Sustainable Campus" project, its operational lines and the results achieved in energy and water consumptions and in sustainable mobility." --- This paper presents a case study. However, the purpose of conducting this case study is not at all clear. What is research problem and hypothesis of this study?

I suggest re-writing of abstract and conclusion to clearly highlight research problem, hypothesis and contributions.

  • Thank you for the relevant question. Aiming at addressing it, as suggested, the following text was added:

At the beginning of the abstract:

This paper addresses the research problem of determining the level of energy and water savings that may be obtained in university campuses without significant investments, based mainly on “surgical” technical and organizational measures.”

  • At the beginning of the conclusions:

“The presented case study has contributed to positively respond the initial research question of determining if relevant energy and water savings may be obtained in university campuses without significant investments, based mainly on ‘surgical’ technical and organizational measures. The data obtained until 2018 were used, when the savings have practically entered a plateau. It was considered that, from that point on, further significant savings required significant investments. In this sense, the authors have prepared and submitted an application to program POSEUR, the Portuguese Operational Program for Sustainability and Efficiency in the Use of Resources, financed by the EU. The application was successful and has allowed obtaining a financing of 5 million euros for a broad program with significant sustainable measures at Alameda campus that have been implemented since 2019.”

Round 2

Reviewer 1 Report

Dear authors,

thank you for your anwers. The paper can be accepted for publishing.

Kind regards

Author Response

The authors thank the reviewer for appreciating the work developed at IST and for contributing forits improvement.

Kind regards,

João Gomes Ferreira

Reviewer 2 Report

The manuscript can be accepted. However, it is advisable to increase the number of references as currently only 14 papers are cited. Increase it to at least 25. There is a lot of research in this area. suggestions:

  1. An integrated approach to achieving campus sustainability: assessment of the current campus environmental management practices
  2. Sustainable campus: engaging the community in sustainability
  3. Students’ Assessment of Campus Sustainability at the University of Dammam, Saudi Arabia 
  4. True Green and Sustainable University Campuses? Toward a Clusters Approach
  5. Improving The Accuracy Of Building Energy Simulation Using Real-Time Occupancy Schedule And Metered Electricity Consumption Data
  6. Life Cycle Assessment of Flat Roof Technologies for Office Buildings in Israel
  7. Effect of evapotranspiration on performance improvement of photovoltaic-green roof integrated system

Author Response

Thank you for the suggestions of relevant references that will surely improve our manuscript.  Besides the suggested seven references, other five were added. The paper has now 26 references, which is in accordance with the suggestion of having at least 25. The new references were placed in the text where they make sense, and all references were renumbered according to the order of appearance.

Suggested references:

Habib M. Alshuwaikhat, Ismaila Abubakar, An integrated approach to achieving campus sustainability: assessment of the current campus environmental management practices, Journal of Cleaner Production, Volume 16, Issue 16, 2008, Pages 1777-1785, ISSN 0959-6526, https://doi.org/10.1016/j.jclepro.2007.12.002.

Too, L. and Bajracharya, B., "Sustainable campus: engaging the community in sustainability", International Journal of Sustainability in Higher Education,2015,  Vol. 16 No. 1, pp. 57-71. https://doi.org/10.1108/IJSHE-07-2013-0080

Abubakar, I.R.; Al-Shihri, F.S.; Ahmed, S.M. Students’ Assessment of Campus Sustainability at the University of Dammam, Saudi Arabia. Sustainability 2016, 8, 59. https://doi.org/10.3390/su8010059

Sonetti, G.; Lombardi, P.; Chelleri, L. True Green and Sustainable University Campuses? Toward a Clusters Approach. Sustainability 2016, 8, 83. https://doi.org/10.3390/su8010083

Prashant, A.; Yang, J.; Cheong, D.; Sekhar, C. Improving The Accuracy Of Building Energy Simulation Using Real-Time Occupancy Schedule And Metered Electricity Consumption Data, ASHRAE Annual Conference, Long Beach, USA, 2017.

Pushkar, S. Life Cycle Assessment of Flat Roof Technologies for Office Buildings in Israel. Sustainability 2016, 8, 54. https://doi.org/10.3390/su8010054

Gupta, S.; Anand, P.; Kakkar, S.; Sagar, P.; Dubey,A. Effect of evapotranspiration on performance improvement of photovoltaic green roof integrated system. International Journal of Renewable Energy, Vol. 12, No. 1, January - June 2017

Additional references:

Velazquez, L.; Munguia, N.; Platt, A.; Taddei, J. Sustainable university: what can be the matter?, Journal of Cleaner Production, Volume 14, Issues 9–11, 2006, Pages 810-819, ISSN 0959-6526, https://doi.org/10.1016/j.jclepro.2005.12.008.

Faghihi, V.; Hessami, A. R.; Ford, D. N. Sustainable campus improvement program design using energy efficiency and conservation, Journal of Cleaner Production, Volume 107, 2015, Pages 400-409, ISSN 0959-6526, https://doi.org/10.1016/j.jclepro.2014.12.040.

Thomashow, M. “The Nine Elements of a Sustainable Campus”. The MIT Press, Cambridge, Massachusetts, 2014.

Amaral, A.R.; Rodrigues, E.:  Gaspar, A.R., Gomes, A. A review of empirical data of sustainability initiatives in university campus operations, Journal of Cleaner Production, Volume 250, 2020, 119558, ISSN 0959-6526, https://doi.org/10.1016/j.jclepro.2019.119558.

Fonseca, P., Moura, P., Jorge, H. and de Almeida, A. Sustainability in university campus: options for achieving nearly zero energy goals, International Journal of Sustainability in Higher Education, Vol. 19 No. 4, 2018, pp. 790-816. https://doi.org/10.1108/IJSHE-09-2017-0145

 Kind regards,

João Gomes Ferreira